# Intradermal Treatment with a Hyaluronic Acid Complex Supplemented with Amino Acids and Antioxidant Vitamins Improves Cutaneous Hydration and Viscoelasticity in Healthy Subjects

**DOI:** 10.3390/antiox13070770

**Published:** 2024-06-26

**Authors:** Gabriel Siquier-Dameto, Pere Boadas-Vaello, Enrique Verdú

**Affiliations:** 1Dameto Clinics International, 1171 VC Badhoevedorp, The Netherlands; info@dametoclinics.com; 2Research Group of Clinical Anatomy, Embryology and Neuroscience (NEOMA), Department of Medical Sciences, University of Girona, E-17003 Girona, Catalonia, Spain; pere.boadas@udg.edu

**Keywords:** hyaluronic acid, skin aging, antioxidant vitamins, skin hydration, skin viscoelasticity, skin rejuvenation, cutaneous microcirculation, mesotherapy, aesthetic medicine

## Abstract

Intradermal injection of bioactive compounds is used to reduce the effects of aging skin. The aim of this work is to study the response of facial injection of a hyaluronic acid complex supplemented with amino acids and antioxidant vitamins on skin rejuvenation. A total of 40 healthy adult subjects were recruited to whom this complex was injected into the facial skin, three consecutive times every two weeks. Together with assessing the degree of skin hydration, the level of skin microcirculation, wrinkles, skin color, and skin biomechanical parameters were evaluated. Using the GAIS scale, the degree of satisfaction of the participants was assessed. At 42 days (D42), there was an 11–12% increase in skin hydration and viscoelasticity, a 23% increase in skin density, a 27% increase in skin microcirculation, and a significant lightening and whitening of skin color, but without causing changes in skin wrinkles. A value between 1 and 3 on the GAIS scale was observed between 70 and 92% of the participants, and 87% of subjects found their skin more beautiful, 85% would recommend this treatment, and more than 50% found their face rejuvenated. In summary, the intradermal treatment tested suggests skin rejuvenation, with a good degree of safety.

## 1. Introduction

Aesthetic medicine includes noninvasive and minimally invasive cosmetic procedures to improve physical appearance and patient satisfaction. Patients who are seeking nonsurgical facial aesthetic procedures do not suffer from any disease and enjoy excellent health. However, they want a minimally invasive procedure that allows controlling the normal effects of aging, especially in the outermost part of the body, which is the skin and its attached structures (hair and nails) [1]. The human skin is constantly exposed to internal (gene mutation, cellular metabolism, hormonal factors) and external (ultraviolet radiation, pollutants, chemical, toxins) stimuli that influence skin function with aging, manifesting as wrinkling, sagging, loss of elasticity, dryness, a reduced barrier integrity, and thinning of the epidermis [2,3,4,5].

Mesotherapy consists of stimulating the biorejuvenation of the skin through minimally invasive epidermal injections or intradermal injections of bioactive substances. The injected products are released long-term into the surrounding tissues, with a deposition effect [6,7]. Among skin rejuvenation products through mesotherapy, hyaluronic acid (HA) has a relevant role in skin hydration due to its high capacity to attract water molecules [8,9], and for this reason it is one of the most-used biologically active substances to promote skin rejuvenation [10,11,12,13,14]. It should be noted that high molecular weight hyaluronic acid has functional hydroxyl groups that can absorb reactive oxygen species (ROS) [15,16,17]. High molecular weight hyaluronic acid interacts with the CD44 receptor, activating intracellular cascades that regulate the redox state and ROS levels of cells [15,16,18]. Polyanionic HA molecules chelate Fe^2+^ and Cu^2+^ ions, which are required in Fenton’s reaction [15,19]. The Fenton reaction is the reaction primarily between iron and hydrogen peroxide, generating a hydroxyl radical, which is highly reactive and highly toxic to living cells. Copper ions can also interact with hydrogen peroxide to generate hydroxyl radicals. In the absence of these ions, the hydroxyl radical cannot be generated [15,20,21]. All of this evidence shows the antioxidant properties of hyaluronic acid, which when applied to the skin also favor the rejuvenation of this tissue.

In mesotherapy treatment, it is common to use hyaluronic acid supplemented with biomolecules including antioxidant, vitamins, and amino acids, aiming to observe the enhancement and/or synergistic effect of these biomolecules with hyaluronic acid [22,23,24,25,26] to minimize the effects of skin aging. These studies show that these combination treatments with hyaluronic acid partially reverse the signs of skin aging, improving the brightness, turgor, and texture of the skin. No adverse events have been reported [22,23,24,25]. However, there are studies that show that facial mesotherapy based on hyaluronic acid with vitamin complexes does not provide any skin benefits [26]. Here, we have studied the efficacy and safety of the complex CELLBOOSTER^®^ Lift (CBL) in healthy adult subjects of both sexes, who presented moderate signs of skin aging. CBL is composed of high molecular weight hyaluronic acid, not cross-linked, and mechanically stabilized by shear deformation and simultaneous pressure, and supplemented with amino acids (arginine, glycine, lysine, proline, valine), biotin, riboflavin, and antioxidative vitamins (C and E). Based on the manufacturer’s instructions, this hyaluronic acid complex is indicated to reduce moderate skin depression, and to improve dermis redensification, skin hydration, and skin microcirculation. In the present study, CBL was applied to healthy adult subjects with signs of skin aging, such as mild or moderate wrinkles, reduced skin laxity, dry skin, and dull skin on the face. Several skin quality measurements were obtained through different instruments, including skin elasticity, density, dryness, microcirculation, wrinkles, and color/homogeneity. Clinical improvement was also evaluated, as well as subject and investigator satisfactions. The results indicate that intradermal CBL treatment significantly improves skin hydration, skin viscoelasticity, and skin microcirculation. Furthermore, a significant aesthetic improvement was reported both by the investigators and by the subjects and most of the subjects were satisfied with the results on their skin after CBL treatment. Regarding safety, all safety parameters analyzed indicate that the safety of CBL appeared to be good. Based on these results, it can be concluded that the treatment with CELLBOOSTER^®^ Lift is safe and efficient in skin hydration. It makes the skin smooth again and it increases skin microcirculation, which together decreases the skin effects of aging, giving a rejuvenated skin appearance.

## 2. Materials and Methods

### 2.1. Experimental Design, Recruiting of Health Subjects, and Mesotherapy Treatment with CBL

This study was designed by the research team of the NEOMA research group at the University of Girona and the clinical trials were carried out at the Eurofins Dermscan Pharmascan international center, based in Lyon (France), under the execution and supervision of the Dr. Gabriel Siquier-Dameto. Eurofins Dermscan Pharmascan (France) is a benchmark in clinical trials of cosmetics and pharmaceutical products. This center specializes in clinical studies and in interventional and noninterventional biomedical research. With ISO 2001 certification, Eurofins Dermscan Pharmascan offers methodologies to assess the safety and efficacy of dermocosmetic products.

The present study consisted of recruiting healthy adult subjects of both sexes who underwent intradermal injections with CBL and assessing the effectiveness of improving the degree of hydration, microcirculation, wrinkles, color, and biomechanical parameters of the skin at 2 and 8 weeks from the last treatment. The degree of improvement in the treatment on the GAIS scale was also assessed, and the degree of satisfaction with the results were obtained using a questionnaire, both by the study subjects and by the medical specialists responsible for the treatment. Finally, CBL safety through the study period was evaluated.

In this study, 40 healthy adult subjects (36 women and 4 men) were recruited following the inclusion/exclusion criteria indicated in Table 1. Every two weeks the subjects underwent a product injection session, and this regimen was repeated three times. In each of the sessions, 3 mL of CBL was administered to the entire face except for the forehead, using the micropapula technique (Figure 1). CELLBOOSTER^®^ Lift (CBL) comprises mechanically stabilized hyaluronic acid (6 mg/mL) with amino acids (arginine, glycine, lysine, proline, valine) and vitamins (riboflavin, biotin, sodium ascorbyl phosphate, tocopherol) in phosphate buffer solution (pH: 6.8–7.6), developed by Suisselle SA (Yverdon-les-Bains, Switzerland) with the patented CHAC technology.

In this research, the principles of the Declaration of Helsinki (1964) were followed, and it is in accordance with the EN ISO 14155:2020 standard [27] and the EU regulation 2017/745 of 5 April 2017. The ethics committee called “Committee de Protection des personnes Ile de France; Address: Hôpital Hôtel Dieu—1, place du Par-vis Notre dame 75004 Paris, France” approved the procedure, and all recruited subjects were informed in writing and verbally of the clinical investigation (information about the research device, its nature and duration, the conditions for carrying out the research, etc.) before the realization of any specific investigational procedure. No subject was included before having signed the consent form, written in an understandable language. The consent process took place during the screening visit.

### 2.2. Evaluation of the Effectiveness Parameters of the Treatment with CBL

To assess the level of hydration in the stratum corneum of the skin, the Corneometer^®^ CM 825 device (Courage-Khazaka electronic GmbH, Köln, Germany) was used, performing a measurement of 10–20 µm (stratum corneum) in depth to exclude the influence of deeper skin layers. The measuring principle is based on capacitance measurement. The surface of the measurement head modifies its electrical capacity according to the humidity level of the measured zone [28,29,30]. Each measurement was an average of five acquisitions. Hydration rate was expressed in arbitrary units (AU). An increase in Corneometer^®^ value characterized a skin more hydrated.

In the evaluation of the rheological properties of the skin, including measurements of biological extensibility and elasticity variations, the MPA 580 Cutometer^®^ (Courage-Khazaka electronic GmbH, Köln, Germany) apparatus was used. The Cutometer@ measurement is carried out by suctioning the skin using negative pressure and this mechanically deforms the skin tissue. The negative pressure generated by the device attracts the skin towards the probe, which releases it again after a certain time. In the probe, the degree of penetration is determined by the Cutometer principle based on a noncontact optical measurement system, where opposing prisms project a beam of light from the transmitter to the receiver. The light intensity changes depending on the depth of penetration into the skin. The skin’s resistance to negative pressure (firmness) and its ability to return to the initial position (elasticity) are presented as real-time curves (penetration depth in mm/time). With these curves, a variety of interesting parameters associated with the elastic and viscoelastic properties of the skin surface are calculated [31,32,33]. Cutaneous skin elasticity measurement was performed with a 6 mm probe, with one cycle of measurement and a 350 mbar constant pressure. Suction and relaxation times were 3 s. Each measurement was an average of two acquisitions. In this study, cutaneous firmness, elasticity, tonicity, and suppleness parameters were studied.

Measurements related to skin density were performed directly in vivo using the Dermascan^®^ C 2D high-frequency ultrasound (Cortex Technology, Aalborg, Denmark). The device consists of an ultrasound machine where a piezoelectric ceramic piece emits an ultrasound beam, which is partially reflected by the separation interface of two media with different ultrasonic impedances. The 20 MHz ultrasound probe is applied directly to the skin, and the use of a contact gel allows homogeneous diffusion of the signal. The Dermascan^®^ C 2D apparatus is connected to a PC computer containing the appropriate software provided by Cortex Technology [34,35,36]. This methodology facilitates two-dimensional visualization of the skin, at the epidermis and dermis level, with 13 mm of penetration. In the image acquisition, echogenic connective tissue appears colored (from yellow to green), and nonechogenic tissues (water) are seen in black. The studied parameter is the proportion of the nonechogenic surface to the total surface analyzed (data expressed in %). A decrease of this ratio characterizes a skin with a higher density.

Cutaneous microcirculation measurements were performed using a PeriFlux 5000^®^ Laser Doppler System equipped with thermostatic probes #457 enabling simultaneous measurements of blood perfusion and cutaneous temperature (Perimed AB, Stockholm, Sweden). The laser–Doppler technique is based on velocimetric measurement of red blood cells moving in superficial microvessels. The laser light beam is guided by an optical fiber to the measurement area and diffuses into a small volume of tissue. Part of the energy is absorbed by the tissue and part is reflected by fixed or mobile structures. The light reflected by moving structures, basically erythrocytes, changes frequency according to the spatial Doppler principle or Doppler effect, while the light reflected by fixed structures does not suffer the Doppler effect. The incident wavelength laser is 780 nm, and the depth of measurement is 0.5 to 1 mm. The degree and change in wavelengths are proportional to the number and speed of erythrocytes, but are not related to the direction of erythrocyte movement. The information captured by the receiving fiber is transformed into an electronic signal that is treated and analyzed. Measurements are expressed in perfusion units (P.U.) and are calculated according to the following formula: Perfusion (or capillary flux) = Number of moving cells × Average speed [37,38].

Cutaneous microrelief was directly evaluated in vivo using the PRIMOS^®^ 3D fringe projection system (Canfield Scientific, Inc.; Parsippany–Troy Hills, NJ 07054, USA). The PRIMOS Clinical Research System was used for the investigation and documentation of the microstructure and wrinkles of the skin. The system allows for measuring the roughness of the skin, wrinkles, and the formation of nodules. The technique consists of using a fringe projector that projects interference fringes on the cutaneous surface, and with a camera to capture the cutaneous area on which the interference fringes are projected. Using the appropriate software, the image captured by the camera is analyzed where the projected fringe network is distorted by the irregularities of the skin surface, and this allows for calculating the height of each point, and by extension, the depth of the wrinkles [39,40].

Skin color parameters that include: (i) degree of skin pigmentation from the individual typological angle (ITA), the individual whitening angle (IWA), and the H76 parameter or degree of skin color homogeneity; (ii) skin luminosity from a luminescence map; (iii) degree of saturation in the skin color; (iv) and the relationship between lightness and degree of saturation, were obtained from skin photographs captured with the 2D Visia^®^ CR camera (Canfield Scientific, Inc.; Parsippany–Troy Hills, NJ 07054, USA) and processed using appropriate software [41,42,43].

### 2.3. Evaluation of the Aesthetic Improvement and the Degree of Satisfaction of the Subject after Treatment with CBL, and Evaluation of the Degree of Satisfaction of the Medical Practitioner with the CBL Injection

Aesthetic improvement after CBL injection was assessed using the global aesthetic improvement scale (GAIS). GAIS is a subjective rating of improvement in treatment results at a given time point compared to pretreatment based on the following score: (1) *exceptional improvement*—excellent corrective result and optimal cosmetic result of the injectable in this subject; (2) *much improvement*—the appearance is significantly improved, but it is not completely optimal, so a new treatment would slightly improve the result; (3) *improvement*—improvement in appearance compared to the initial condition, but new treatment is required and indicated; (4) *unaltered or unchanged*—compared to the initial condition, the appearance remains the same, without significant changes; (5) *worse*—compared to the original or initial situation, the appearance has worsened [44].

The degree of satisfaction of the subject who has received the treatment with CBL was carried out by means of a questionnaire with the following questions and/or considerations: (i) Has this intervention been good for me?; (ii) Am I satisfied with the results obtained?; (iii) I consider that my face has been rejuvenated with the treatment; (iv) I consider that my skin looks more hydrated after the treatment; (v) I consider that my skin is with less redness after the treatment; (vi) I consider that my skin looks less pigmented after the treatment; (vii) I consider that my skin looks more beautiful after the treatment; (viii) I consider that my skin is firmer after the treatment; (ix) Would you say that the results obtained seem natural?; (x) Would you recommend this treatment that you have undergone to a friend or a family member? Each of these ten items was rated on a scale of (1) not at all, (2) a little, (3) a lot, and (4) totally; the subject had to mark a value from this scale for each of the items in the questionnaire.

The degree of satisfaction of the medical practitioner with the CBL injection was assessed with the following questionnaire: (i) ease of injection; (ii) ease of product positioning; (iii) ease of vial manipulation; (iv) ease of turbidity detection. Each of these four items was rated on a scale of (1) very satisfied, (2) satisfied, (3) indifferent, (4) dissatisfied, and (5) very dissatisfied; the medical practitioner had to mark a value from this scale for each of the items in the questionnaire, after each subject and each injection session.

### 2.4. Safety Assessment

Throughout the study period, the safety of CBL, the injection device, and the subjects was evaluated by collecting several parameters, including the (i) assessment of *injection site reactions* (ISR) by the medical practitioner at the start of the study, immediately after each CBL injection and at different time days of the study, and by the subject who received the treatment for 2 weeks after each injection session. The following ISRs were tabulated by maximum severity and duration for each CBL injection session: (ii) collection of adverse events (AEs) during the study, such as *adverse medical events*, unintentional illness or injury, or adverse clinical signs in study subjects, users, or others, related or unrelated to the medical device under investigation; *adverse device effect* on the use of the investigational medical device, including insufficient or inappropriate instructions for use or any malfunction of the investigational medical device, and use error or intentional misuse of the investigational medical device research product; *deficiency of the medical device* in relation to its identity, reliability, durability, quality, ease of use, safety, or performance, including errors of use and/or malfunction, and inadequacy of the information provided by the manufacturer (including labeling); and (iii) collection of *serious adverse events* (SAEs) including death, serious deterioration in the health of the subject induced by an illness or injury that may be life-threatening, permanent deterioration of a body structure or body function, hospitalization or prolongation of the period of hospitalization of the subject, surgical and/or medical intervention as prevention of injuries or illnesses that endangers the life of the subject, chronic illness.

The severity of adverse events (AEs) was classified on a three-point scale: (1) *mild*—mild discomfort, without effects on the subject’s daily activities, and without the need to take concomitant treatments; (2) *moderate*—some discomfort with some effects on the daily activities of the subject, and the possibility of taking concomitant treatments; and (3) *severe*—great discomfort of the subject that affects daily activities, and necessary concomitant treatment. On the other hand, in relation to ISRs, the following parameters were analyzed: (i) erythema/redness, (ii) pain/sensitivity, (iii) hardening/induration, (iv) edema/swelling, (v) protuberances/bumps, (vi) hematoma/bruise, (vii) pruritus/itching, (viii) loss and/or increase in color/pigmentation, and (ix) other. These parameters were assessed on a scale of none (0), mild (1), moderate (2), and severe (3). The scoring of the ISRs was performed by the medical practitioner, but also by the subject during the following 14 days after each injection. These subject scores were reviewed by the medical practitioner at the next injection session. The maximum intensity and duration of each ISR were computed and analyzed.

### 2.5. Statistical Analyses

The data from the present study were collected during the 6 weeks from the start of the first injection session with CBL. The results are shown as mean ± standard deviation. The Shapiro–Wilk test was used to verify the normality of the groups, and the Student’s *t* test was used to compare the parameter. Likewise, in the analysis by gender (men vs. women) the t test was used. In all cases there was an alpha risk of 5%. The SPSS 25.0 program for Windows was used in this statistical analysis.

## 3. Results

### 3.1. Cutaneous Hydration Rate Evaluated with the Corneometer@ Apparatus

An optimal hydration rate evaluated with the Corneometer^®^ is between 60 and 80 arbitrary units (A.U.) [29,45], although a value equal to or greater than 40–45 AU is considered a sufficient degree of hydration [46,47]. The baseline value (D7) of the subjects in this study was 44.81 ± 8.66 AU, which suggests that these subjects presented slightly dry skin. This parameter increased significantly after CBL treatment (D42) (Figure 2), up to 50.38 ± 9.36 AU. This significant increase was about 11%. In other words, these results suggest that the CBL treatment improves the hydration rate of the skin by 11%.

When analyzing the hydration rate by gender, men showed values of 35.43 ± 12.35 AU and 45.57 ± 7.25 AU at D7 and D42, respectively, while in women it was 45.83 ± 7.74 AU and 50.77 ± 9.48 AU at D7 and D42, respectively. In men, there were no significant differences in this parameter between days 7 and 42 (*p* > 0.05), while in women there were significant differences between days 7 and 42 (*p* < 0.05). This analysis by gender suggests that CBL treatment increases the degree of skin hydration, with significant differences being observed in women but not in men.

### 3.2. Skin Biomechanical Parameter Evaluated Using the Cutometer^®^ Apparatus

Table 2 shows the results obtained on various biomechanical parameters of the skin using the Cutometer^®^ apparatus at baseline (D7) and after treatment (D42) with CBL.

No significant differences were observed with CBL treatment for the parameters of skin flexibility, firmness, tone, and elasticity (*p* > 0.05), but there were significant differences after CBL treatment regarding skin plasticity, and especially for the viscoelasticity of the skin, which increased by 12.5%. Thus, these results indicate that CBL treatment improves skin viscoelasticity by 12.5%, which may be associated with a similar increase in the degree of skin hydration.

Analyzing the skin viscoleasticity parameter by gender, the values observed on D7 and D42 were 0.21 ± 0.03 and 0.28 ± 0.08 in men and 0.26 ± 0.28 and 0.24 ± 0.05 in women, with no significant differences (*p* > 0.05) observed in this parameter between D7 and D42 in men and women.

### 3.3. Skin Density Assessed with the Dermascan^®^ C 2D High-Frequency Ultrasound Device

The results of skin density evaluated with the Dermascan^®^ apparatus are shown in Figure 3. With respect to the basal value (D7), the treatment with CBL (D42) significantly increased the density of the skin (proportion of nonechogenic tissue). Treatment with CBL improves skin density by 22.6%.

The analysis of this parameter according to the gender of the recruited subjects observed that at D7 and D42, the value (%) was 22.50 ± 6.46 and 29.67 ± 6.03, respectively, in men, and 22.73 ± 10.23 and 29.31 ± 12.74, respectively, in women. In women but not in men, significant differences (*p* < 0.05) in this parameter were observed between days 7 and 42.

### 3.4. Cutaneous Microcirculation Assessed with the Laser–Doppler System Technique

Treatment with CBL causes a significant increase in skin microcirculation (Figure 4). The average value at D7 was 41.83 and at D42 it was 57.55, with an improvement of 27.3%. These results suggest that CBL treatment significantly improves blood flow through the skin.

When this parameter was analyzed based on the gender of the recruited subjects; at D7 and D42 the values were 76.19 ± 39.71 and 103.9 ± 21.71 in men, and 38.01 ± 24.87 and 53.79 ± 37.30 in women, on the respective days. In women but not in men, significant differences (*p* < 0.05) in this parameter were observed between the two evaluation days.

### 3.5. Skin Wrinkle Parameters Assessed with the PRIMOS^®^ 3D Fringe Projection System

Using Primos^®^, various parameters of crow’s feet antiwrinkle microrelief were evaluated after treatment with CBL. No significant difference was observed in these parameters with the treatment compared to the baseline situation (Table 3). As the *p* values are close to 0.05, these results suggest that more CBL treatments are needed to obtain statistical significance.

Analysis of these parameters after two days of evaluation based on the gender of the recruited subjects showed that there were no significant differences (*p* > 0.05) between both days in men and women. The average roughness was 23.97 ± 1.86 and 26.25 ± 0.33 at D7 and D42, respectively, in men, and 22.37 ± 6.55 and 22.93 ± 6.89 at D7 and D42, respectively, in women. On the other hand, the average height of the roughness was 185.0 ± 32.67 and 203.6 ± 29.89 at D7 and D42, respectively, in men, and 170.9 ± 55.35 and 177.0 ± 59.48 at D7 and D42, respectively, in women. Finally, the maximum height of the roughness profile was 128.2 ± 18.62 and 136.0 ± 8.64 at D7 and D42, respectively, in men, and 116.3 ± 31.52 and 120.6 ± 36.16 at D7 and D42, respectively, in women.

### 3.6. Skin Color Parameters

Table 4 shows the results obtained from the different parameters evaluated regarding skin color in the baseline situation (D7) and after treatment with CBL (D42).

CBL treatment caused significant changes in skin color, and it specifically caused a 2.92% increase in lightness, a 10.71% increase in ITA, a 2.69% increase in IWA, luminance increased by 2.92%, and the luminance–saturation ratio increased by 4.97%. In contrast, the treatment significantly decreased redness by 8.33%, the ratio of yellow components by 4.42%, the H76 parameter by 3.30%, and the degree of saturation by 5.39%.

For days 7 and 42, by gender of the subjects recruited, the analysis of the previous parameters of skin color after treatment with CBL showed some significant differences. In relation to lightness, the values obtained in men are 62.51 ± 3.93 and 65.28 ± 2.29 at D7 and D42, respectively, and in women they are 68.80 ± 2.5 and 70.79 ± 2.36 at D7 and D42, respectively. Significant differences (*p* < 0.001) were observed in women but not in men. The values of the redness parameter were 21.74 ± 2.23 and 20.47 ± 3.48 at D7 and D42 in men, respectively, and in women 17.07 ± 2.11 and 15.61 ± 1.72 to D7 and D42, respectively. For this parameter, significant differences (*p* < 0.01) were also observed in women but not in men. The yellow component values were 24.72 ± 4.59 and 22.47 ± 3.58 in men at D7 and D42, respectively, and 23.01 ± 2.22 and 22.09 ± 2.06 in women at D7 and D42, respectively. For this parameter, no significant differences were observed in either men or women. In relation to the degree of pigmentation (ITA), the values were 27.11 ± 11.14 and 34.09 ± 7.67 at D7 and D42 in men, respectively, and 39.01 ± 5.70 and 42.83 ± 5.13 at D7 and D42 in women, respectively. Significant differences (*p* < 0.01) were observed only in women. The whiteness ratio (IWA) was 61.66 ± 4.35 and 64.64 ± 1.97 at D7 and D42 in men, respectively, and 67.17 ± 2.36 and 68.83 ± 2.14 at D7 and D42 in women, respectively. Significant differences (*p* < 0.01) were also observed only in women. Color homogeneity (H76) was 6.79 ± 1.24 and 6.09 ± 0.67 at D7 and D42 in men, respectively, and 6.09 ± 0.67 and 5.50 ± 0.68 in women at D7 and D42, respectively. No significant differences were observed in either men or women. The color saturation parameter was 31.66 ± 3.65 and 29.58 ± 1.73 at D7 and D42 in men, respectively, and in women 27.27 ± 2.16 and 25.83 ± 2.11 at D7 and D42, respectively. Significant differences (*p* < 0.01) were observed only in women. Finally, the luminance–saturation ratio in men was 54.73 ± 6.15 and 58.75 ± 3.34 at D7 and D42, respectively, and in women it was 64.0 ± 3.95 and 67.12 ± 3.78 at D7 and D42, respectively. Significant differences (*p* < 0.001) were only observed in women.

### 3.7. Global Aesthetic Improvement Scale (GAIS) Assessed by the Medical Practitioner and by the Subject

The results of the aesthetic improvement with the CBL treatment evaluated by the GAIS scale are shown in Table 5. The medical practitioner indicated that the aesthetic improvement had improved/much improved by 82.5% (improved 62.5%, much improved 20%), whereas the subject indicated that improved/much improved was 80% (improved 50%, much improved 30%). However, no change after CBL treatment was 17.5% by medical practitioner and 20% by subjects. Overall, the improvement of the aesthetic aspect with the CBL treatment was about 80–82%.

The GAIS analysis by gender of the recruited subjects is shown in Table 6.

One hundred percent of the recruited men indicate that their appearance had improved after treatment with CBL. In the recruited women, 2.6% indicated that their appearance had very much improved after CBL treatment, 28.9% indicated that it had much improved, 47.4% that it had improved, and 21% that it had not changed. The impression of medical practitioner in relation to the improvement in men treated with CBL was 34% much improvement, 33% improvement, and 33% no change, while for women it was 18.4% much improvement, 65.8% improvement, and 15.8% unchanged.

### 3.8. Degree of Satisfaction of the Subject Who Had Received the Treatment with CBL

The results of the satisfaction questionnaire passed to the subjects treated with CBL are shown in Table 7. At D42, the overall satisfaction of the subjects was good—90.0% of the subjects responded “a little”, “a lot”, or “totally” to the following questions: “Has this intervention been good for me?”, “I am satisfied with the results obtained”, “I consider that my skin looks more hydrated after the treatment”, and “I consider that my skin is firmer”. Furthermore, a total of 87.5% of subjects found their skin more beautiful, 85% would recommend this treatment, and more than 50% found their face rejuvenated, less red, and less pigmented.

### 3.9. Degree of Satisfaction of the Medical Practitioner with the CBL Injection

The degree of satisfaction indicated by the medical practitioners regarding the CBL injection was 100% in all of the items in the questionnaire from all of the CBL injection sessions. That is, these physicians were very satisfied with the ease of injection, ease of product positioning, ease of vial manipulation, and ease of turbidity detection.

### 3.10. Normality Values of the Evaluated Parameters and Changes between Men and Women

In some of the previous paragraphs, normality values for some of the parameters evaluated were already included, such as the degree of skin hydration assessed with the Corneometer^®^ method. It should be noted that, for this parameter, subjects were sought who initially had low values of this parameter, so that treatment with CBL could produce notable changes.

Regarding the degree of skin viscoelasticity using the Cutometer ^®^ method, it was determined that normal elasticity is between 0.5 and 1.5 in Caucasian subjects between 18 and 70 years of age [48]. In another study, it was established that basal skin elasticity is 0.27 ± 0.02 in Caucasian subjects between 40 and 60 years old [49]. In Caucasian subjects between 23 and 35 years old, this parameter is 0.202 ± 0.07 [50]. The range of this parameter assessed in the facial skin of Caucasian subjects aged 40 to 55 years is between 0.28 and 0.47 [51]. Despite this variability among studies, it could be indicated that the degree of skin elasticity in the facial skin of Caucasian subjects aged between 40 and 55–60 years is at least 0.27–0.28. The baseline value (D7) observed in the subjects recruited in the present study was 0.21 ± 0.04 (range: 0.17–0.25), indicating that they had slightly low skin elasticity. It should be noted that subjects who had values slightly lower than normal were sought to better discern the changes generated with the treatment.

In healthy Caucasian subjects between 46 and 65 years old, microcirculation assessed by the Doppler technique ranges between 14 and 58 PU [52]. In the skin of the arm of Caucasian subjects between 27 and 57 years old, it was determined that microcirculation assessed by the Doppler technique gives values in a range from 16 to 35 PU [53]. In the interdigital region of the skin, cutaneous microcirculation assessed by the Doppler technique gave values of 3–7 PU in Caucasian subjects between 40 and 66 years old [54]. In healthy Caucasian subjects between 27 and 51 years of age, microcirculation assessed by the Doppler technique gave values in the ranges of 8–14 PU and 57–130 PU in the forearm and hand, respectively [55]. In the present study, microcirculation evaluated with the Doppler technique in Caucasian volunteers between 35 and 55 years old was 41.83 ± 28.48 PU, with an approximate range of 13–70 PU.

In healthy subjects between 19 and 73 years old, the evaluation of the Ra (roughness) parameter of wrinkles in facial skin using the PRIMOS device gave values between 16 and 39 AU [56]. In Caucasian women aged 30 to 60 years, the value of the Ra parameter assessed with the PRIMOS device at the level of the facial skin was 22.71 ± 4.40 AU [57]. In adult Asian women, the value of Ra and Rz in the facial skin, assessed with the PRIMOS method, was 26.39 ± 1.85 AU and 121.41 ± 8.91 AU, respectively [58]. In healthy volunteers aged 35 to 75 years, the Ra and Rz parameters assessed with the PRIMOS device were 31.9 AU and 172.2 AU in the periorbital skin, respectively; and 40.1 AU and 216.8 AU, in the mesolabial skin, respectively [59]. In healthy volunteers aged 20 to 60 years, on the skin of the forearm, the Ra and Rz parameters assessed with the PRIMOS device gave values in the ranges 25–50 AU and 125–200 AU, respectively [60]. Altogether, in all of these studies carried out on healthy subjects aged between 20 and 70 years, using the PRIMOS device to determine the Ra and Rz parameters, the values of these parameters are approximately 20–40 and 120–200 AU, respectively. In the present study, the values of these two parameters are in those ranges.

Through skin photographs captured with various cameras and coupled software, different degrees in human skin color were established. For an intermediate skin color with an ITAº value of 28–41, the lightness parameter (L*) gave a value (mean ± SEM) of 63.3 ± 0.4, the redness parameter (a*) of 7.4 ± 0.5, and the yellow component parameter (b*) of 18.7 ± 0.5 [61]. In European Caucasian subjects, the values of the L* parameter range between 60 and 73 [62]. In a study carried out with university students from the United States, the values of the parameters L*, a*, and b* were 69.04 ± 11.79, 13.09 ± 1.68, and 25.51 ± 2.60, respectively [63]. In Caucasian subjects, the values of L*, a*, b*, ITAº, and IWA were approximately 45, 25, 30, and 50 AU, respectively [43]. In Caucasian women aged 20 to 60 years, facial skin L*, a*, and b* values were 62–68, 10–15, and 15–22 AU, respectively [64]. These findings from these different studies suggest that the normal values of the parameters L*, a*, and b* are approximately 63–68, 13–25, and 18–30 AU, respectively.

In relation to the differences between men and women at D42, significant differences were found in the following parameters: cutaneous microcirculation (*p* < 0.05), L* parameter (*p* < 0.001), a* parameter (*p* < 0.001), ITAº parameter (*p* < 0.01), IWA parameter (*p* < 0.01), skin color saturation (*p* < 0.019), and luminance–saturation ratio (*p* < 0.001). For the other parameters analyzed, there were no observed significant differences (*p* > 0.05) between men and women. Most of these parameters with significant differences between men and women are related to the appearance of skin color. Treatment with CBL in women makes the skin tone (ITAº) light, while in men it is intermediate. The skin whiteness ratio in women is greater than in men after treatment with CBL. The appearance of the skin color of women is more lightness (L*) than in men, with less redness (a*) than in men, after treatment with CBL. The saturation of the skin color of women after treatment with CBL is lower than in men, with a higher luminance–saturation ratio in women than in men after treatment with CBL. The lower cutaneous microcirculation in the skin of women compared to men after treatment with CBL may also contribute to skin color that gives a more rejuvenated appearance.

It should be noted that these differences between men and women should be taken with caution, since the number of men who were recruited was much lower than the number of women, even though an open call was made, and people who wanted to participate voluntarily appeared for the study. In the future, more studies should be conducted in men on the effect of CBL to compare the results observed between men and women.

### 3.11. Security Analysis

Injection site reactions (ISRs) were classified as none, mild, moderate, and severe. Table 8 indicates the rate of ISRs that occurred during the entire study ranged from 100% to 92.7%.

As shown in Table 9, ISRs were assessed by medical practitioners as follows. There were no severe reactions after the injections. Most reactions were mild including lumps/bumps (ranging from 95.1% to 85.4%), redness/erythema (ranging from 97.6% to 78%), edema (ranging from 46.3% to 10%), pain/tenderness (ranging from 50% to 26.8%), bruising/hematoma (ranging from 7.5% to 2.6%), itching (2.4%), and discoloration/pigmentation (2.4%). Only a few moderate reactions were reported, including redness/erythema (ranging from 17.1% to 5%), lumps/bumps (12.2%), pain/tenderness (ranging from 2.5% to 2.4%), and edema (2.4%).

None of the ISRs were reported by the medical practitioner as “severe”, regardless of the endpoint assessment.

Table 10 suggests that there was a small difference between the percentages assessed by medical practitioners and patients.

Patients were asked to rate the ISR with a severity score. Table 11 shows no severe reactions, few moderate reactions, and some mild reactions; most patients had no reaction.

In any case, most of the ISRs lasted 1–3 days, some 4–7 days, and only a few 8–14 days, as indicated in Table 12.

Patients reported all adverse events that occurred during the entire study. No serious adverse event was reported. Only 12.2% were adverse events related or potentially related to the device injection (mild hematoma in three subjects and moderate pain in two subjects, and a stye on the left eye in one subject).

A summary of reported adverse events is shown in Table 13.

## 4. Discussion

In healthy adult subjects 35–55 years of age, of both sexes (mainly women) and with signs of dry skin, mild-moderate wrinkles, and reduced skin laxity, treatment with CBL consisting of three sessions of cutaneous injections of the product every 2 weeks, significantly improved the degree of skin hydration, skin viscoelasticity, and skin microcirculation. The skin complexion radiance, including skin color parameters, also improved. In addition, a significant aesthetic improvement was reported both by the medical practitioners and by the subjects, and most of the subjects were satisfied with the results on their skin. Regarding safety, all ISRs and ADEs reported during the study were expected and resolved. The safety of CELLBOOSTER^®^ Lift (CBL) appeared to be good. All of these findings suggest that the treatment with CELLBOOSTER^®^ Lift (CBL) is safe and efficient in giving an aesthetic appearance of rejuvenation, as it improves skin parameters towards a youthful skin trend.

Skin hydration is essential for maintaining the skin’s mechanical barrier, preventing the entry of microorganisms. The degree of water retained by hyaluronic acid in the dermis and epidermis influences the hydration of the skin, and the maintenance in the degree of hydration of the skin is dependent on the stability of the granular layer of the epidermis [9,65]. Hyaluronic acid is a molecule that captures and releases water, capable of binding 1000 times its volume in water. This water absorption property allows the compound to adequately hydrate the different layers and/or strata of the skin, both at the dermal and epidermal level [9,65,66]. The predominant hyaluronic acid in vivo has molecular weights greater than 1000 KDa, and this is called high molecular weight HA (HMW-HA) [67], although in vivo there is also hyaluronic acid with molecular weights between 20 and 1000 KDa, called low molecular weight hyaluronic acid (LMW-HA), which frequently originates from enzymatic cleavage of HMW-HA [68]. The molecular weight of HA influences the biological effects of this compound, thus HMW-HA has anti-inflammatory, antiangiogenic effects, and immunosuppressive activity, while LWM-HA induces inflammation, immune response, and angiogenesis [69]. Likewise, the degree of penetration of the HA in skin also depends on the molecular weight. LMW-HA easily penetrates the skin, especially the epidermis, reaching very superficial layers of the dermis, while HMW-HA is unable to penetrate the skin, remaining limited only to the stratum corneum of the epidermis [9,70]. On the other hand, changes in the levels of water retained by hyaluronic acid are not related to the molecular weight of the compound [71]. All of these findings suggest that high molecular weight hyaluronic acid should be injected into the skin, since it has a low capacity for cutaneous penetration, but it is a compound of the extracellular matrix that retains high amounts of water, therefore its cutaneous application favors the skin hydration, despite the fact that it is easily degraded by various extracellular proteases, generating low molecular weight hyaluronic acid, which still maintains an optimal degree of skin hydration. Like CELLBOOSTER^®^ Glow [72], CELLBOOSTER^®^ Lift (CBL) consists of HMW-HA, not cross-linked and stabilized by simultaneous mechanical forces of shear and pressure deformation. Therefore, the cutaneous injection of CBL favors the hydration of the skin, owing to the high hygroscopic capacity of high molecular weight hyaluronic acid. Various previous studies have shown that dermal injection of hyaluronic acid favors the degree of hydration, and this gives a rejuvenated appearance to the skin [73,74,75], very similar to what is found in the present study.

The dermal application of CBL causes a 12.5% increase in the viscoelasticity of the skin, which may be due to several factors. Firstly, the degree of hydration of the skin can influence the viscoelasticity of this tissue. In this sense, a relationship was found between these two parameters, so that the greater the cutaneous hydration, the greater the degree of viscoelasticity of the skin [76]. Studies in which skin creams were applied observe greater skin viscoelasticity associated with a higher degree of skin hydration [77,78]. Likewise, spraying fine particles of water on dry skin improves the viscoelasticity of this skin [79]. Taken together, all of these findings suggest a relationship between the viscoelasticity of the skin and the degree of hydration in this tissue, and in the context of the present study, this increase in hydration is due to the application of hyaluronic acid containing the CBL compound injected into the skin; consequently, this induces an increase in the viscoelasticity of the skin. Secondly, inflammatory processes can induce a decrease in the viscoelasticity of the skin. An increase in plasmatic levels of inflammation markers correlates with a lower degree of viscoelasticity of the skin [80]. Ultraviolet radiation (UVA, UVB) on the skin, which causes inflammation [81], triggers a decrease in skin viscoelasticity [82]. It is well known that ultraviolet radiation also generates oxygen free radicals in the skin, which favors skin aging [83,84]. Oxygen free radicals and/or reactive oxygen species (ROS) (the oxygen free radicals are part of the reactive oxygen species, [85]), induce degradation of elastic fibers [86] and hinder the polymerization of elastic fibers [87]. ROS causes the degradation of collagen fibers via activation of metalloproteinases [88,89,90]. ROS also triggers a reduction in collagen synthesis by dermal fibroblasts [91]. All of these findings suggest that inflammatory mediators and ROS that appear in the skin because of exposure of it to environmental aggressors (e.g., ultraviolet radiation, atmospheric pollutants) cause changes in the composition of the extracellular matrix that ultimately influence on the viscoelasticity of the skin. Vitamin C has an antioxidant role, acting as a scavenger of oxygen free radicals. This vitamin also promotes collagen synthesis by dermal fibroblasts. Vitamin E also acts as a free radical scavenger [92]. At the level of the epidermis, vitamin C favors the synthesis of collagen by keratinocytes and decreases the lipid peroxidation of these epidermal cells. Keratinocytes have the capacity to accumulate high amounts of vitamin C, which protects them from the effects of oxygen free radicals [93]. In addition to the antioxidant role of vitamin E, it has a role in preventing inflammatory processes in the skin [94] and it is a protective vitamin for dermal fibroblasts, reducing the expression of proinflammatory cytokines [95]. On the other hand, HMW-HA has anti-inflammatory effects, preventing the production of proinflammatory chemical mediators and the activation of extracellular matrix metalloproteinases. Likewise, HMW-HA has antioxidant effects acting as a free radical scavenger [15,96]. All of these findings suggest that vitamins C and E, as well as high molecular weight hyaluronic acid (HMW-HA), exert anti-inflammatory and antioxidant effects, thereby favoring the generation of an environment that improves the viscoelasticity of the skin. In the context of the present study, it should be noted that the CELLBOOSTER^®^ Lift compound contains both vitamins and high molecular weight hyaluronic acid. Thirdly, essential (e.g., valine, lysine, proline) and nonessential (e.g., glycine) amino acids influence the homeostasis of collagen levels and other components of the extracellular matrix in the skin. Some of these amino acids favor collagen synthesis by fibroblasts [97,98,99,100]. Likewise, the combination of these amino acids (e.g., arginine, glycine, proline) is important to re-establish collagen levels in ultraviolet-irradiated skin [101]. The mixture of hyaluronic acid and amino acids regulates the expression of extracellular matrix genes in fibroblast cultures, specifically alanine and valine together with hyaluronic acid, promote the synthesis of elastin and collagen [102]. The mixture of glycine, alanine, proline, valine, leucine, and lysine also regulates gene expression and favors the synthesis of elastin, fibronectin, and collagen (types I and IV) in cultured human fibroblasts [103]. All of this evidence indicates that a certain group of amino acids influences the synthesis of components of the extracellular matrix by skin fibroblasts, and together with this they can affect skin viscoelasticity. The CBL compound used in the present study contains valine, lysine, proline, glycine, and arginine—amino acids that influence the homeostasis of the extracellular matrix of the skin. Together, all the components of CBL contribute directly or indirectly to the homeostasis of the components in the extracellular matrix of the skin, and skin viscosity is improved with this contribution.

CBL treatment also improves skin density by 22.6%. The concept of skin density is related to the activity of fibroblasts and keratinocytes, and to the content of components in the extracellular matrix of the skin, which together cause changes in the thickness of the epidermis and dermis [36,104,105,106]. Hyaluronic acid stimulates the proliferation of fibroblasts [107] and the synthesis of collagen and other molecules in the extracellular matrix by these cells [102,108,109]. It also stimulates the proliferation of keratinocytes in the epidermis [110]. Vitamin C promotes collagen synthesis by dermal fibroblasts [111,112,113], and essential and nonessential amino acids also promote the synthesis of collagen and other extracellular matrix compounds by skin fibroblasts [97,98,99,100,101,102,103]. Together, hyaluronic acid, vitamin C, and amino acids influence the deposition of extracellular matrix components and the proliferation of dermal fibroblasts that affect the thickness of the skin and with it the parameter of skin density. All of these molecules are found in the chemical composition of the CBL compound applied to the subjects of the present study.

A 27.3% increase in cutaneous microcirculation was observed after cutaneous treatment with CBL. Neoangiogenesis is promoted by vascular endothelial growth factor (VEGF), and this increases skin microcirculation. VEGF is synthesized and secreted by both keratinocytes in the epidermis and fibroblasts in the dermis [114,115]. In vitro, it has been observed that hyaluronic acid favors the secretion of VEGF by fibroblasts, which increases microcirculation [116]. An increase in cutaneous microcirculation implies a greater blood perfusion of the skin and together with it a greater supply of nutrients and oxygen, while the removal of carbon dioxide, metabolites, and inflammatory factors is favored. Basal levels of blood perfusion in the cutaneous microcirculation of the face are significantly higher than those observed in other body regions [117]. The constant exposure of the skin of the face to aggressive agents (e.g., ultraviolet radiation, atmospheric pollutants) can cause changes in the cutaneous microcirculation. In this context, preclinical studies have shown that ultraviolet radiation causes a significant decrease in VEGF expression [118]. These results are contradictory with most studies that observed an overexpression of VEGF after ultraviolet irradiation [119,120,121]. However, ultraviolet radiation in transgenic mice that overexpress VEGF induces skin damage associated with degradation of the dermal matrix and enhanced vascularization [122]. Likewise, VEGF contributes to the increase in vascularization after ultraviolet irradiation [123]. All of these findings show a relationship between an increase in VEGF due to ultraviolet radiation accompanied by angiogenesis, and therefore greater microcirculation, which may explain this increase in microcirculation in facial skin. The cutaneous application of a hyaluronic acid-based compound that regulates VEGF expression by skin cells can further increase this basal microcirculation, as observed in the present study. On the other hand, treatment with vitamin C and E improves skin microcirculation in patients undergoing hemodialysis, since both vitamins reduce oxidative stress [124]. Likewise, treatment with ascorbic acid (vitamin C) and L-arginine improves skin microcirculation in patients with chronic renal failure [125]. Taken together, these findings suggest that some components of the CBL compound (e.g., vitamins C and E, and amino acids such as arginine) are involved in the improvement of cutaneous microcirculation that was observed in the present study.

CBL treatment causes significant changes in all evaluated parameters associated with skin color. Treatment with CBL significantly increases the lightness (luminosity) of the skin. It is well known that young skin is characterized by having a greater overall luminosity than aged skin [126]. Despite this, there is no universal definition of skin luminosity, since it is a multifactorial parameter that depends on radiance, illumination, brightness, and transparency, among others. Skin color is also affected by skin microcirculation (redness) and accumulation of yellowish pigments [127,128]. There is evidence to suggest that an increase in skin microcirculation decreases skin redness [129,130]. It should be noted that CBL treatment increases skin microcirculation and significantly reduces the ratio of yellow component of skin color. Skin color changes with aging, becoming redder and darker in older subjects. Likewise, the yellow color of the skin also increases at advanced ages [131,132]. On the other hand, ITAº permits the classification of skin tone. The skin tone according to the individual typology angle (ITAº) is as follows: very light (ITAº greater than 55), light (ITAº between 41 and 55), intermediate (ITAº between 28 and 41), tan (ITAº between 10 and 28), brown (ITAº between minus 30 and plus 10), and dark (ITAº less than minus 30) [133]. Treatment with CBL produces an increase of about 11% in ITAº, which causes the skin to change from an intermediate color (D7) to a light color (D42) on the previous scale. In addition, the individual whitening angle (IWA) is a parameter that evaluates the whiteness of the skin [43]. Treatment with CBL causes an increase of approximately 3% in this parameter, suggesting that CBL favors the generation of whiter skin. Parameter H76 permits the identification of color dispersion compared to the average value [134]. CBL treatment causes a significant 3% decrease in this parameter, which suggests that it prevents color dispersion, and this translates into greater skin whiteness. Likewise, CBL treatment causes a reduction in color saturation, which results in an attenuation of skin color, and causes an increase in the luminance–saturation ratio, which causes lower saturation in skin colors but to have a shine. Taken together, all of the changes related to skin color tend to show rejuvenated skin.

Using the GAIS for evaluating the aesthetic improvement after CBL treatment, this improvement ranged between 70 and 92.5% of the cases, that is, a GAIS value between 1 and 3 was found in 70–92.5% of the cases. In a previous study, a GAIS value between 1 and 2 was observed in 90% of the cases treated with low and high molecular weight hyaluronic acid in combination with plasma [135]. A similar percentage for aesthetic improvement has been described after treatment with TEOSYAL RHA^®^ 4 [136]. The degree of aesthetic improvement with CBL is equal to or slightly higher than that observed with other treatments with hyaluronic acid. Regarding the degree of satisfaction with the CBL treatment, 87.5% of subjects found their skin more beautiful; 85% would recommend this treatment and more than 50% found their face rejuvenated, less red, and less pigmented. In a previous study, about 75% of the subjects were satisfied with the treatment, which consisted of the injection of polymethylmethacrylate (PMMA) into the dorsal skin of the hand [137]. The degree of satisfaction was 84 to 94% of the subjects who administered the HA dermal filler (Juvéderm(^®^) VOLBELLA(^®^)) to the lips [138]. In another study, 87.9% of the patients treated with Profhilo (IBSA) on the face showed a high degree of satisfaction with the treatment received [139]. The level of patient satisfaction with the treatments received depends on the type of treatment and the area treated, but the degree of satisfaction indicated by patients who received facial treatment with CBL is like those who also received facial treatment with Profhilo.

In relation to the safety analysis of the procedure used in the present study, the different parameters analyzed show that on day 42 (D42) all patients showed no signs of reaction to the CBL injection. The signs described were mostly mild in severity. The moderate signs with the greatest presence were redness and cutaneous erythema, pain, edema, and lumps/bumps after the first injection with CBL. Most of these signs lasted between 1 and 3 days after CBL injection. Finally, a total of 30 adverse events (AEs) were reported in 46.3% of subjects in the present study, and most of these AEs could be considered mild in severity. In summary, the safety of CELLBOOSTER^®^ Lift appeared to be good. In a previous study where Profhilo (IBSA) was administered to the face, it was shown that the adverse events were mild in severity, and their duration did not exceed 3 days [139]. The trial where full face augmentation was performed using Tissuefill indicates that the incidence of complications was low [140]. Despite the scarcity of studies assessing the safety of the procedure, the results obtained with CELLBOOSTER^®^ Lift are like those previously described. Actually, as there is a trend in natural results and repeated treatments, it will have a cumulative result improving wrinkles because of better tissue structure and elasticity.

There are no previous studies that assess the effect of skin treatment with hyaluronic acid according to gender. There are few studies that assess the effect of skin treatment with antioxidants according to gender. In this sense, the oral administration of antioxidant vitamins and minerals affects the incidence of skin cancer differentially between women and men, being higher in women with antioxidant treatment [141]. There is evidence that women use sunscreen creams more than men [142]. Likewise, in adult volunteers from India, it has been observed that the percentage of women who consume antiaging and antiwrinkle cosmetic products is 18.75%, while in men it is 12.5%, and the percentage sunscreen consumption among women and men was 22.08% and 25.83%, respectively [143]. On the other hand, it is known that sex hormones differentially affect the physiology of the skin, influencing men and women differently [144,145,146]. Knowing this gender-dependent skin physiology can help develop better-targeted cosmetic treatments between men and women, with the intention of promoting antiaging treatments that prevent, repair, and protect the skin with age, and by extension allow skin to have a better appearance, smoother, clearer, and younger-looking skin [144]. In part, the treatment tested in the present study had differential effects between men and women, observing that CBL induced a lighter skin tone in women compared to men, with a greater proportion of skin whiteness, greater lightness, and less redness. All of this provides a perception of rejuvenation of women’s skin after testing of the treatment, compared to men. There is evidence to suggest that hyaluronic acid [14], antioxidant vitamins (e.g., vitamin C) [147], and amino acids mixed with hyaluronic acid [148,149,150] rejuvenate human skin. The compound used in the present study contains all of these skin rejuvenating components.

## 5. Conclusions

In healthy adult volunteers, facial injection of 3 mL/session of CBL in three injection sessions with an interval of 2 weeks between each session causes significant increases in the degree of skin hydration, skin density with greater viscoelasticity, increased skin microcirculation, and enhanced lightening and whitening of skin color. The degree of satisfaction of the participants is high, and 87% of them say that their skin is more beautiful. In conclusion, facial treatment with CBL promotes a rejuvenated appearance of the skin, which should be attributed to its antioxidant components, mainly hyaluronic acid and vitamins (C, E, B2, and biotin).

## Figures and Tables

**Figure 1 antioxidants-13-00770-f001:**
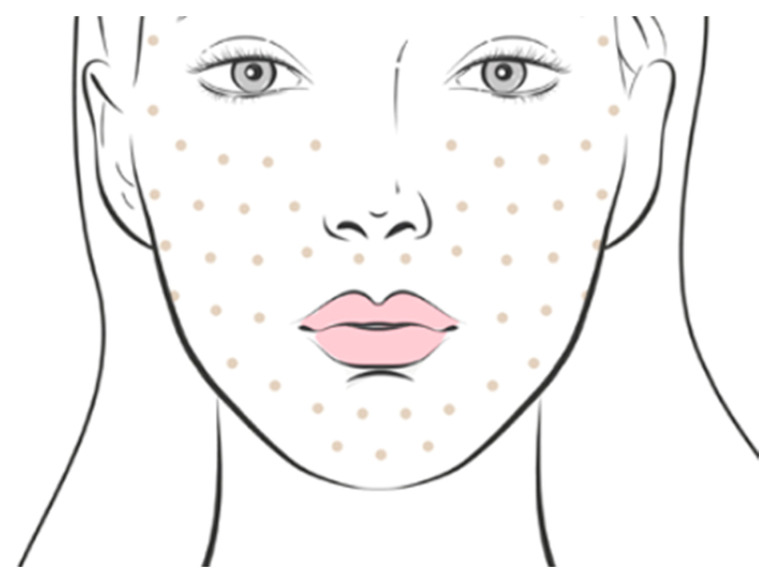
Micropapule technique used for facial administration of CBL. This technique consists of administering small volumes of CLB at different points (papules) distributed throughout the area of the face where the product needs to be applied.

**Figure 2 antioxidants-13-00770-f002:**
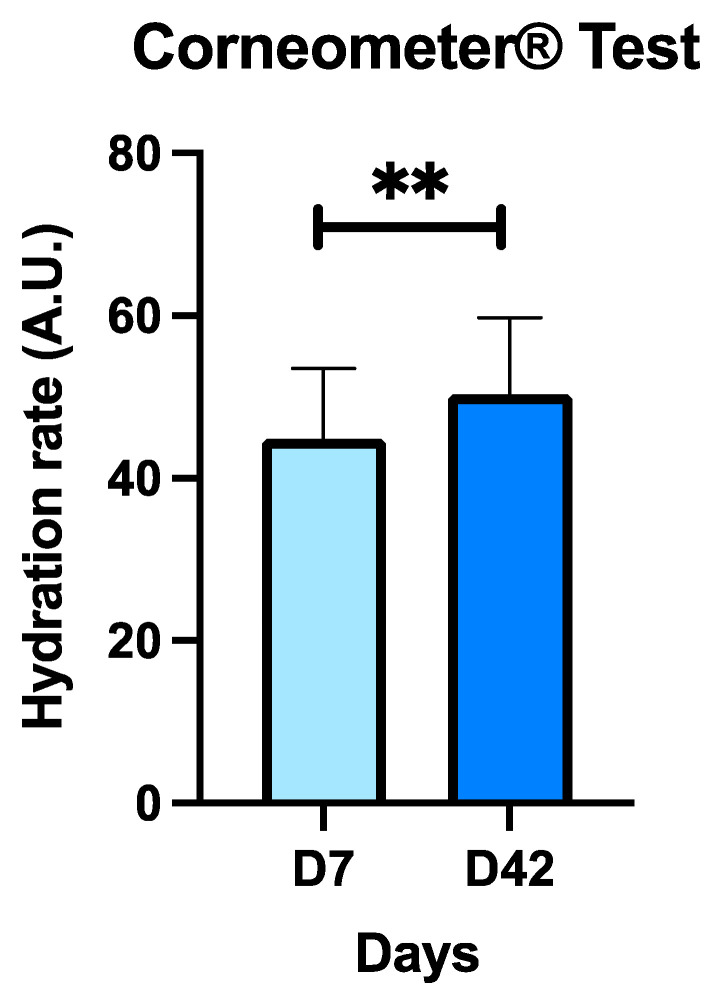
Results of the hydration rate of the skin of the subjects in the baseline situation (D7) before and after treatment with CBL (D42) using the Corneometer^®^ apparatus. Values are mean ± SD. Forty subjects (n = 40) were analyzed. ** *p* < 0.01 with respect to baseline (D7).

**Figure 3 antioxidants-13-00770-f003:**
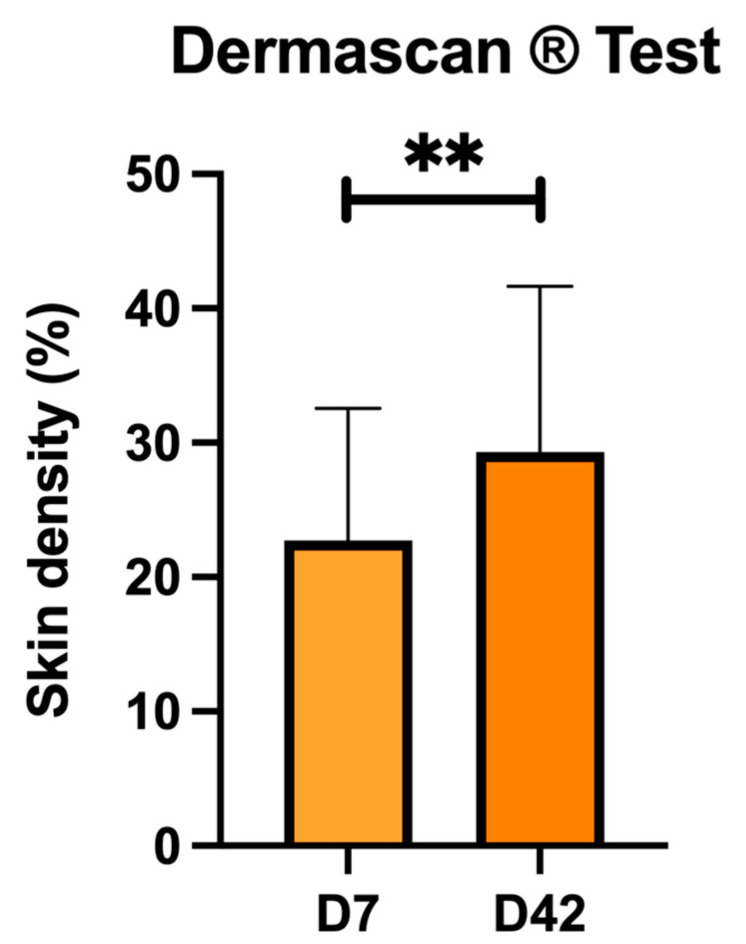
Results of the skin density of the subjects in the baseline situation (D7) and after treatment with CBL (D42) using the Dermascan^®^ device. The parameter evaluated is the proportion of nonechogenic tissue (skin density %). Values are mean ± SD. Forty subjects (n = 40) were analyzed. ** *p* < 0.01 with respect to baseline (D7).

**Figure 4 antioxidants-13-00770-f004:**
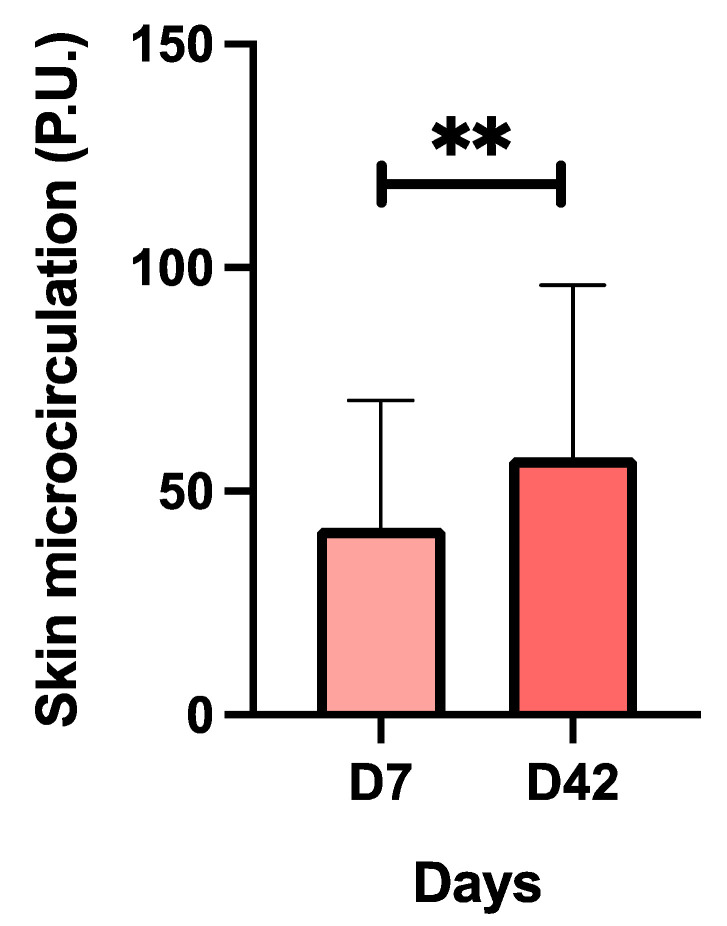
Results of the skin microcirculation of the subjects in the baseline situation (D7) and after treatment with CBL (D42) using the laser–Doppler technique. Values are mean ± SD. Forty subjects (n = 40) were analyzed. ** *p* < 0.01 with respect to baseline (D7).

**Table 1 antioxidants-13-00770-t001:** Inclusion and exclusion criteria used in the recruitment of subjects in the present study.

Inclusion Criteria
Healthy subject Men and women, aged between 35 and 55 years Caucasians, with phototype II to III Patients with signs of facial skin chronoaging, dull skin, facial dryness, laxity of the skin, and presence of wrinkles but not severe. Subject with degree of skin hydration on cheekbones <60 AU (assessed with Corneometer^®^) Subject with psychological capabilities to understand the information associated with the study, and with the capabilities to give written informed consent. Subject who has given free, express, and informed consent; who is registered in a social security regime; and who is willing to refrain from performing other facial aesthetic procedures on the entire face for the entire duration of the study. Women of childbearing potential must have a negative pregnancy test at the time of inclusion and must use a medically accepted method of contraception for at least 12 weeks before the start of the study and throughout the study.
**Exclusion Criteria**
○ Women who are lactating, pregnant, or planning to become pregnant during the study, and women with menopause for at least 1 year in perimenopause, without hormonal treatment. ○ Subject with intensive exposure to UV rays or sunlight during the previous month, and/or plans to do so during the study. ○ Subject with tattoos, scars, moles, or too much hair, as well as anything else on the face that could interfere with the study, based on the opinion of the researcher. ○ Subject who, in the last year before the start of the study, received injections of resorbable filler products, laser or ultrasound-based treatment, dermabrasion, deep chemical peel, or another ablative procedure on the face, as well as facial surgery in the last two years. ○ Subject who in the previous 9 months received treatment with botulinum toxin on the face, such as one who had received mesotherapy products on the face in the previous 3 months, and one who had a superficial peel in the previous 2 months, or a superficial exfoliation on the face. ○ Subject with subcutaneous retention structures on the face (mesh, steps, gold thread), as well as one who had received injections of permanent–semipermanent fillings on the face, and one who uses cosmetics with alpha-hydroxy acids. ○ Subject with alcoholism (more than 10 glasses of wine/day) and smoking (more than 10 cigarettes/day), as well as one with a BMI > 30, one with unstable body weight, or one who plans to follow a dietary regimen during the study. ○ Subject with depression and/or ongoing, uncontrolled or recently recovered psychiatric disorders (<6 months), or any other disorder that represents a risk to the health of the subject in the study, and that may have an impact on the study evaluations, as well as subject deprived of his/her liberty by administrative or judicial decision, and one who is under guardianship. ○ Subject with serious, ongoing, and uncontrolled diseases (e.g., malignancy or history of malignancy, type I diabetes, liver failure, renal failure, lung/heart disease, neoplasia, malignant blood disease, tumor, HIV, or other major disease), as well as a subject with recurrent porphyria, coronary insufficiency, ventricular rhythm disorders, severe hypertension, obstructive cardiomyopathy, or hyperthyroidism. ○ Subject with a known history or suffering from an autoimmune disease and/or immunodeficiency, with multiple allergies, a history of anaphylactic shock and evolving allergic pathologies, as well as a subject with coagulation and/or homeostasis disorders, or on anticoagulant treatment or treatment that may interfere with the hemostasis process, during the previous month and during the study. ○ Subject with any systemic disease (acute and/or chronic), in the previous year, likely to interfere with the measured parameters or to put the subject to an undue risk, as well as a subject with any skin disease including cutaneous inflammatory or infectious processes, abscess, unhealed wound, or a cancerous or precancerous lesion on the face, and a subject predisposed to keloids or hypertrophic scarring or pigmentation disorders. ○ Subject having history of hypersensitivity to the antiseptic solution, to lidocaine and/or prilocaine or local anesthetics of amide type or one of the excipients of EMLA 5% cream, as well as a subject having history of allergy or hypersensitivity to one of the components of the tested device. ○ Subject undergoing a topical or systemic treatment: anti-inflammatory medication during the previous 2 weeks and during the study; antihistaminics during the previous 3 days; immunosuppressors and/or corticoids during the 4 previous weeks and during the study; retinoids during the 6 previous months and during the study; and subjects receiving any treatment that, in the opinion of the clinical investigators, may interfere with test results or put the subject to undue risk.

**Table 2 antioxidants-13-00770-t002:** Results of the biomechanical parameters of the skin evaluated with the Cutometer^®^ device.

Parameter	D7	D42	
Skin flexibility	0.79 ± 0.22	0.78 ± 0.22	*p* = 0.64
Firmness	0.95 ± 0.25	0.97 ± 0.25	*p* = 0.61
Skin plasticity	0.16 ± 0.04	0.18 ± 0.05	*p* = 0.01
Skin tonicity	0.49 ± 0.17	0.50 ± 0.18	*p* = 0.30
Elasticity	0.61 ± 0.07	0.63 ± 0.09	*p* = 0.15
Viscoelasticity	0.21 ± 0.04	0.24 ± 0.06	*p* = 0.0038

**Table 3 antioxidants-13-00770-t003:** Results of skin wrinkle parameters evaluated by Primos^®^.

Parameter	D7	D42	
Average of roughness	22.49 ± 6.32	23.18 ± 6.67	*p* = 0.0566
Average of height of the roughness	116.43 ± 30.35	121.74 ± 35.01	*p* = 0.1394
Maximum height of the roughness profile	171.83 ± 53.68	179.17 ± 57.99	*p* = 0.0529

**Table 4 antioxidants-13-00770-t004:** Results of the various parameters of skin color.

Parameter	D7	D42	
Lightness	68.41 ± 3.01	70.40 ± 2.71	*p* < 0.0001
Redness	17.42 ± 2.40	15.97 ± 2.21	*p* < 0.0001
Yellow component ratio	23.14 ± 2.38	22.12 ± 2.12	*p* < 0.0001
ITA	38.09 ± 6.71	42.17 ± 5.66	*p* < 0.0001
IWA	66.77 ± 2.82	68.57 ± 2.33	*p* < 0.0001
H76	5.73 ± 0.70	5.54 ± 0.68	*p* < 0.0001
Saturation	27.60 ± 2.49	26.11 ± 2.26	*p* < 0.0001
Luminance–saturation ratio	63.33 ± 4.69	66.48 ± 4.27	*p* < 0.0001

**Table 5 antioxidants-13-00770-t005:** Results of the global aesthetic improvement scale (GAIS).

GAIS Grades	Medical Practitioner	Subject
Very much improved	0%	0%
Much improved	20%	30%
Improved	62.5%	50%
No change	17.5%	20%
Worse	0%	0%

**Table 6 antioxidants-13-00770-t006:** Results of the global aesthetic improvement scale (GAIS) by gender.

	Men		Women	
GAIS Grades	MP	Subj	MP	Subj
Very much improved	0%	0%	0%	2.6%
Much improved	34%	0%	18.4%	29.9%
Improved	33%	100%	65.8%	47.4%
No change	33%	0%	15.8%	21.0%
Worse	0%	0%	0%	0%

MP: Medical practitioner; Subj: Subject.

**Table 7 antioxidants-13-00770-t007:** Results of the satisfaction questionnaire passed to the CBL-treated subjects.

Questions	Not at All	A Little	A Lot	Totally
(i)	10%	35%	40%	15%
(ii)	10%	37.5%	50%	2.5%
(iii)	25%	52.5%	22.5%	0%
(iv)	10%	42.5%	40%	7.5%
(v)	30%	45%	20%	5%
(vi)	37.5%	30%	25%	7.5%
(vii)	12.5%	42.5%	32.5%	12.5%
(viii)	10%	60%	25%	5%
(ix)	7.5%	10%	37.5%	45%
(x)	15%	35%	22.5%	27.5%

Questions: (i) Has this intervention been good for me?; (ii) Am I satisfied with the results obtained?; (iii) I consider that my face has been rejuvenated with the treatment; (iv) I consider that my skin looks more hydrated after the treatment; (v) I consider that my skin is with less redness after the treatment; (vi) I consider that my skin looks less pigmented after the treatment; (vii) I consider that my skin is looks more beautiful after the treatment; (viii) I consider that my skin is firmer after the treatment; (ix) Would you say that the results obtained seem natural?; (x) Would you recommend this treatment that you have undergone to a friend or a family member?

**Table 8 antioxidants-13-00770-t008:** Proportion of subjects presenting at least one sign at each visit as assessed by the medical practitioner.

Session/Visit		None	At Least Once
I (D0)	After injection	0%	100%
II (D14)	Before injection	92.7%	7.3%
II (D14)	After injection	2.4%	97.6%
III (D28)	Before injection	97.5%	2.5%
III (D28)	After injection	0%	100%
D42		100%	0%

**Table 9 antioxidants-13-00770-t009:** Percentage of subjects who presented the signs in the different visits.

Redness/Erythema					
Session/Visit		None	Mild	Moderate	Severe
I (D0)	After injection	4.9%	78%	17.1%	0%
II (D14)	Before injection	100%	0%	0%	0%
II (D14)	After injection	2.4%	97.6%	0%	0%
III (D28)	Before injection	100%	0%	0%	0%
III (D28)	After injection	2.5%	92.5%	5%	0%
D42		100%	0%	0%	0%
**Pain/Tenderness**					
**Session/visit**		**None**	**Mild**	**Moderate**	**Severe**
I (D0)	After injection	58.5%	39%	2.4%	0%
II (D14)	Before injection	100%	0%	0%	0%
II (D14)	After injection	73.2%	26.8%	0%	0%
III (D28)	Before injection	100%	0%	0%	0%
III (D28)	After injection	47.5%	50%	2.5%	0%
D42		100%	0%	0%	0%
**Induration**					
**Session/visit**		**None**	**Mild**	**Moderate**	**Severe**
I (D0)	After injection	95.1%	4.9%	0%	0%
II (D14)	Before injection	100%	0%	0%	0%
II (D14)	After injection	100%	0%	0%	0%
III (D28)	Before injection	100%	0%	0%	0%
III (D28)	After injection	100%	0%	0%	0%
D42		100%	0%	0%	0%
**Edema**					
**Session/visit**		**None**	**Mild**	**Moderate**	**Severe**
I (D0)	After injection	51.2%	46.3%	2.4%	0%
II (D14)	Before injection	97.6%	2.4%	0%	0%
II (D14)	After injection	97.6%	2.4%	0%	0%
III (D28)	Before injection	97.5%	2.5%	0%	0%
III (D28)	After injection	90%	10%	0%	0%
D42		100%	0%	0%	0%
**Lumps/Bumps**					
**Session/visit**		**None**	**Mild**	**Moderate**	**Severe**
I (D0)	After injection	2.4%	85.4%	12.2%	0%
II (D14)	Before injection	100%	0%	0%	0%
II (D14)	After injection	4.9%	95.1%	0%	0%
III (D28)	Before injection	100%	0%	0%	0%
D42		100%	0%	0%	0%
**Bruising/Hematoma**					
**Session/visit**		**None**	**Mild**	**Moderate**	**Severe**
I (D0)	After injection	97.4%	2.6%	0%	0%
II (D14)	Before injection	95.1%	4.9%	0%	0%
II (D14)	After injection	95.1%	4.9%	0%	0%
III (D28)	Before injection	100%	0%	0%	0%
III (D28)	After injection	92.5%	7.5%	0%	0%
D42		100%	0%	0%	0%
**Itching**					
**Session/visit**		**None**	**Mild**	**Moderate**	**Severe**
I (D0)	After injection	97.6%	2.4%	0%	0%
II (D14)	Before injection	100%	0%	0%	0%
II (D14)	After injection	100%	0%	0%	0%
III (D28)	Before injection	100%	0%	0%	0%
III (D28)	After injection	100%	0%	0%	0%
D42		100%	0%	0%	0%
**Discoloration/pigmentation**					
**Session/visit**		**None**	**Mild**	**Moderate**	**Severe**
I (D0)	After injection	100%	0%	0%	0%
II (D14)	Before injection	100%	0%	0%	0%
II (D14)	After injection	97.6%	2.4%	0%	0%
III (D28)	Before injection	100%	0%	0%	0%
III (D28)	After injection	100%	0%	0%	0%
D42		100%	0%	0%	0%

**Table 10 antioxidants-13-00770-t010:** Proportion of subjects presenting at least one ISR after each injection session, as assessed by the subjects.

Redness/Erythema			
Session/Visit		None	At Least One
I (D0)	After injection	36.6%	63.4%
II (D14)	After injection	34.1%	65.9%
III (D28)	After injection	45%	55%
**Pain/Tenderness**			
**Session/visit**		**None**	**At least one**
I (D0)	After injection	68.3%	31.7%
II (D14)	After injection	56.1%	43.9%
III (D28)	After injection	65%	35%
**Induration**			
**Session/visit**		**None**	**At least one**
I (D0)	After injection	92.7%	7.3%
II (D14)	After injection	92.7%	7.3%
III (D28)	After injection	92.5%	7.5%
**Edema**			
**Session/visit**		**None**	**At least one**
I (D0)	After injection	87.8%	12.2%
II (D14)	After injection	95.1%	4.9%
III (D28)	After injection	92.5%	7.5%
**Lumps/Bumps**			
**Session/visit**		**None**	**At least one**
I (D0)	After injection	51.2%	48.8%
II (D14)	After injection	48.8%	51.2%
III (D28)	After injection	70%	30%
**Bruising/Hematoma**			
**Session/visit**		**None**	**At least one**
I (D0)	After injection	61%	39%
II (D14)	After injection	48.8%	51.2%
III (D28)	After injection	57.5%	42.5%
**Itching**			
**Session/visit**		**None**	**At least one**
I (D0)	After injection	92.7%	7.3%
II (D14)	After injection	85.4%	14.6%
III (D28)	After injection	95%	5%
**Discoloration/pigmentation**			
**Session/visit**		**None**	**At least one**
I (D0)	After injection	90.2%	9.8%
II (D14)	After injection	92.7%	7.3%
III (D28)	After injection	95%	5%

**Table 11 antioxidants-13-00770-t011:** Distribution of ISR severity scores, as assessed by the subject after each injection session.

Redness/Erythema					
Session/Visit		None	Mild	Moderate	Severe
I (D0)	After injection	36.6%	53.7%	9.8%	0%
II (D14)	After injection	34.1%	53.7%	12.2%	0%
III (D28)	After injection	45%	45%	10%	0%
**Pain/Tenderness**					
**Session/visit**		**None**	**Mild**	**Moderate**	**Severe**
I (D0)	After injection	68.3%	24.4%	7.3%	0%
II (D14)	After injection	56.1%	39%	4.9%	0%
III (D28)	After injection	65%	30%	5%	0%
**Induration**					
**Session/visit**		**None**	**Mild**	**Moderate**	**Severe**
I (D0)	After injection	92.7%	7.3%	0%	0%
II (D14)	After injection	92.7%	7.3%	0%	0%
III (D28)	After injection	92.5%	7.5%	0%	0%
**Edema**					
**Session/visit**		**None**	**Mild**	**Moderate**	**Severe**
I (D0)	After injection	87.8%	7.3%	4.9%	0%
II (D14)	After injection	95.1%	4.9%	0%	0%
III (D28)	After injection	92.5%	7.5%	0%	0%
**Lumps/Bumps**					
**Session/visit**		**None**	**Mild**	**Moderate**	**Severe**
I (D0)	After injection	51.2%	36.6%	12.2%	0%
II (D14)	After injection	48.8%	39%	12.2%	0%
II (D14)	After injection	48.8%	39%	12.2%	0%
**Bruising/Hematoma**					
**Session/visit**		**None**	**Mild**	**Moderate**	**Severe**
I (D0)	After injection	61%	29.3%	9.8%	0%
II (D14)	After injection	48.8%	46.3%	4.9%	0%
III (D28)	After injection	57.5%	35%	7.5%	0%
**Itching**					
**Session/visit**		**None**	**Mild**	**Moderate**	**Severe**
I (D0)	After injection	92.7%	4.9%	2.4%	0%
II (D14)	After injection	85.4%	14.6%	0%	0%
III (D28)	After injection	95%	5%	0%	0%
**Discoloration/pigmentation**					
**Session/visit**		**None**	**Mild**	**Moderate**	**Severe**
I (D0)	After injection	90.2%	9.8%	0%	0%
II (D14)	After injection	92.7%	7.3%	0%	0%
III (D28)	After injection	95%	2.5%	2.5%	0%

**Table 12 antioxidants-13-00770-t012:** ISR duration after each injection session, as assessed by the subjects.

Redness/Erythema			
ISR Duration	(D0) after Injection	(D14) after Injection	(D28) after Injection
None	36.6%	34.1%	45%
From 1 day to 3 days	53.7%	53.7%	52.5%
From 4 days to 7 days	7.3%	9.8%	2.5%
From 8 days to 14 days	2.4%	2.4%	0%
More than 14 days	0%	0%	0%
**Pain/Tenderness**			
**ISR duration**	**(D0) after injection**	**(D14) after injection**	**(D28) after injection**
None	68.3%	56.1%	65%
From 1 day to 3 days	26.8%	41.5%	32.5%
From 4 days to 7 days	4.9%	2.4%	2.5%
From 8 days to 14 days	0%	0%	0%
More than 14 days	0%	0%	0%
**Induration**			
**ISR duration**	**(D0) after injection**	**(D14) after injection**	**(D28) after injection**
None	92.7%	92.7%	92.5%
From 1 day to 3 days	4.9%	7.3%	7.5%
From 4 days to 7 days	2.4%	0%	0%
From 8 days to 14 days	0%	0%	0%
More than 14 days	0%	0%	0%
**Edema**			
**ISR duration**	**(D0) after injection**	**(D14) after injection**	**(D28) after injection**
None	87.8%	95.1%	92.5%
From 1 day to 3 days	9.8%	4.9%	7.5%
From 4 days to 7 days	2.4%	0%	0%
From 8 days to 14 days	0%	0%	0%
More than 14 days	0%	0%	0%
**Lumps/Bumps**			
**ISR duration**	**(D0) after injection**	**(D14) after injection**	**(D28) after injection**
None	51.2%	48.8%	70%
From 1 day to 3 days	41.5%	48.8%	27.5%
From 4 days to 7 days	4.9%	2.4%	2.5%
From 8 days to 14 days	2.4%	0%	0%
More than 14 days	0%	0%	0%
**Bruising/Hematoma**			
**ISR duration**	**(D0) after injection**	**(D14) after injection**	**(D28) after injection**
None	61%	48.8%	57.5%
From 1 day to 3 days	17.1%	36.6%	15%
From 4 days to 7 days	17.1%	14.6%	10%
From 8 days to 14 days	4.9%	0%	12.5%
More than 14 days	0%	0%	5%
**Itching**			
**ISR duration**	**(D0) after injection**	**(D14) after injection**	**(D28) after injection**
None	92.7%	85.4%	95%
From 1 day to 3 days	7.3%	14.6%	5%
From 4 days to 7 days	0%	0%	0%
From 8 days to 14 days	0%	0%	0%
More than 14 days	0%	0%	0%
**Discoloration/pigmentation**			
**ISR duration**	**(D0) after injection**	**(D14) after injection**	**(D28) after injection**
None	90.2%	92.7%	95%
From 1 day to 3 days	7.3%	7.3%	2.5%
From 4 days to 7 days	2.4%	0%	0%
From 8 days to 14 days	0%	0%	2.5%
More than 14 days	0%	0%	0%

**Table 13 antioxidants-13-00770-t013:** Summary of AEs.

Description	% AEs	Description	% AEs
Headache/migraine	43.8%	Styes on the left eye	3.3%
Hematoma	10%	Dental pain	3.3%
Anxiety	3.3%	Legs aches	3.3%
Dental abscess	3.3%	Lumbago	3.3%
Influenza status	3.3%	Pain in face and both cheeks	3.3%
Light edema under the right dark circle	3.3%	COVID-19	3.3%
Pain in the cheekbones	3.3%	Cough	3.3%
Muscular soreness	3.3%	Sinusitis	3.3%
**Description**			
Cutaneous AE (No) = 80%; Cutaneous AE (Yes) = 20%			
Action taken with product: None = 100%; Definitive interruption = 0%			
Action taken due to event: None = 16.7%; Corrective treatment = 80%; Other (application of hot compresses soaked in physiological serum) = 3.3%			
SAE: No = 100% ; Yes = 0%			
Relationship with the device: Not related = 80%; Possible = 3.3%; Probable = 6.7%; Causal = 10%			
Relationship with the study methods; Not related = 96.7%; Possible = 3.3%; Probable = 0%; Causal = 0%			

## Data Availability

All data generated or analyzed during this study are included in this published article.

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
