# Peer review of "Intradermal Treatment with a Hyaluronic Acid Complex Supplemented with Amino Acids and Antioxidant Vitamins Improves Cutaneous Hydration and Viscoelasticity in Healthy Subjects"

_antioxidants, 2024, doi:10.3390/antiox13070770_

Round 1

Reviewer 1 Report

Presented manuscript described the interesting results of well planned experiment. However, I am not sure if this kind of experiments/analysis fit to the scope of the Antioxidants – should it be submitted to more ”dermatological” journal.

Despite of above, there are also some questions/suggestions to the authors:

1.       Can't the title be shorter and less detailed?

2.       Why is there such an unequal gender distribution in the study group? Did these 4 men stand out in any way in the results?

3.       The presented results do not have any statistical analysis prepared (standard deviations, significance, perhaps some correlations?)

It cannot be accepted in this form.

Author Response

The reviewers request that the title of the article be shortened, as they consider it excessively long. The authors appreciate this comment, and therefore we have changed the title according to the reviewers' suggestions. The new title is as follows: "Intradermal treatment with a hyaluronic acid complex supplemented with amino acids and antioxidant vitamins improves cutaneous hydration and viscoelasticity in healthy subjects".

Reviewer 1 has indicated that: “Presented manuscript described the interesting results of well-planned experiment. However, I am not sure if this kind of experiments/analysis fit to the scope of the Antioxidants – should it be submitted to more ”dermatological” journal.” The authors appreciate this comment, since it is true that the manuscript can also be submitted to a dermatology journal. However, the manuscript describes the effects on the skin of the application of hyaluronic acid supplemented with amino acids and vitamins, and some of these compounds have antioxidant properties (Kim et al. Food Chem. 2008, 109(4):763-770; Mendoza et al. Mini Rev Med Chem. 2009, 9(13):1479-1488; Pullar et al. Nutrients. 2017, 9(8):866; Njus et al. Free Radic Biol Med. 2020,159:37-43; Jiang. Free Radic Biol Med. 2014, 72:76-90; Olfat et al. Br J Nutr. 2022, 128(10):1887-1895). The authors consider that the scope of this manuscript could be natural and synthetic antioxidants and their relevance to plant, animal and human health and disease, as well as safe antioxidant preservatives for foods, fodder, and cosmetic formulations, two of the scope included in the journal Antioxidants.

Reviewer 1 states that "Why is there such an unequal gender distribution in the study group? Did these 4 men stand out in any way in the results?”. The authors appreciate this comment, but these were the subjects that were recruited. In the analysis of the results, all recruited subjects have been taken. Based on this comment, the results have been reanalyzed, separating between women and men. New results have been added to the manuscript.

Reviewer 1 has indicated that: “The presented results do not have any statistical analysis prepared (standard deviations, significance, perhaps some correlations?)”. The authors accept this comment, but we do not agree with the comment, since the manuscript shows a statistical analysis of the results, with mean and standard deviation, and differential analysis of these results.

The modifications made based on the reviewers' suggestions and comments have been marked in yellow in the modified manuscript. New information based on these suggestions has been marked in green in the modified manuscript.

Reviewer 2 Report

Review report to Intradermal treatment with a hyaluronic acid mechanically stabilized complex supplemented with amino acids and antioxidant vitamins in healthy subjects with signs of skin aging improves cutaneous hydration and viscoelasticity of skin giving an aesthetic rejuvenation appearance“ by Gabriel Siquier-Dameto, Pere Boadas-Vaello and Enrique Verdú.

In this study the authors investigated the effects of a repeated injection of hyaluronic acid in a complex with amino acids and antioxidant vitamins (Cellbooster®Lift) into facial skin. Many different parameters related to improved skin ageing signs were assessed such as skin elasticity, hydration, microcirculation, skin density and skin color. In addition, the patients and the medical practitioners satisfection was scored. The injections were applied to 40 healthy adult subjects during 2 weeks. The results were presented for day 7 and 42. Besides the illustration as bar graphs for hydration, skin density and skin microcirculation, tabels were used to present the results.

The study describes the findings clear and comprehensive. Materials and methods are described in great detail, including the working principle of the cutaneous measurement devices. In contrast, the results are described very brief. The authors could add a few additional sentences to describe the results.

The discussion covers all parameters tested and puts them into context within the existing literature.

There are only a few small comments that could improve the manuscript:

The title is very long. The authors should shorten it. As a suggestion: Intradermal treatment with a hyaluronic acid complex supplemented with amino acids and antioxidant vitamins improves cutaneous hydration and viscoelasticity

In some cases references are missing, i.e.

Line 48: High molecular weight hyaluronic acid interacts with the CD44 receptor, activating intracellular cascades that regulate the redox state and ROS levels of cells. And line 50:…for polyanionic HA molecules chelate Fe/Cu

51: Fenton reaction is the reaction between iron/cooper …. should be copper?

Are the reference values that were cited age matched controls? For the hydration the study cohort seems to start at a very low level. Is this comparable to the published values?

In my opinion it is not necessary to write both, day 7 and D7. Please use the term that is also used in the bar graphs and tables and apply throughout the mansucript.

Fig 3, Dermascan test: The unit is given as % as described in materials and methods but the diagramm shows 0,2..etc. The unit is missing at y-axis. For all parameters it would be helpfull to have a reference value (if available from literature or databases and applicable). For Fig.4 an increase of 37.6 % seems very high, as written in the text. The bars show differences between 45 to 60 PU. Please explain the numbers in the text.

Line 352: „Both the medical practitioner and the subjects who received the CBL treatment indicate that there is an apparent improvement in 50-62.5% of the cases, and that this improvement is important between 20-30% of the cases.“ Please rephrase this sentence as the improvement refer either to the practioner or the patient (i.e. 20 and 30 % judged, respectively). As it is written now, this is not clear to the reader and might be misleading. How do the authors calculate 70-92,5 % from table 5? This is not clear and needs to be explained or rephrased.

Discussion: Carefully check the use of HWM-HA . It seems to got mixed up in the discussion.

Line 532: „well known that ultraviolet radiation also generates oxygen free radicals in“  The authors use the term ROS in the following sentences. Perhaps they could introduce ROS instead of ‚oxygen free radical‘ and use it throughout.

line 620: Luminosity depends on luminosity…... Please rephrase this sentence.

Author Response

The reviewers request that the title of the article be shortened, as they consider it excessively long. The authors appreciate this comment, and therefore we have changed the title according to the reviewers' suggestions. The new title is as follows: "Intradermal treatment with a hyaluronic acid complex supplemented with amino acids and antioxidant vitamins improves cutaneous hydration and viscoelasticity in healthy subjects".

Reviewer 2 indicate: “In some cases references are missing, i.e. Line 48: High molecular weight hyaluronic acid interacts with the CD44 receptor, activating intracellular cascades that regulate the redox state and ROS levels of cells. And line 50:…for polyanionic HA molecules chelate Fe/Cu; 51: Fenton reaction is the reaction between iron/cooper …. should be copper?”. The authors appreciate this comment, more references have been included in the sentences indicated by the reviewer. The wording of Fenton's reaction has also been rewritten, and new bibliographical citations have been added.

Reviewer 2 indicate “Are the reference values that were cited age matched controls? For the hydration the study cohort seems to start at a very low level. Is this comparable to the published values?”. The authors thank the reviewer for their comments. More bibliographic evidence from previous studies has been searched and more information has been added to the manuscript. It has already been specified that the optimal hydration value is between 60-80 AU, but that the minimum hydration value is 40-45 AU. The values ​​obtained in the study are at the minimum of hydration.

Reviewer 2 indicate “In my opinion it is not necessary to write both, day 7 and D7. Please use the term that is also used in the bar graphs and tables and apply throughout the manuscript”. The authors appreciate the comment, and based on this suggestion, the appropriate changes have been made to the manuscript.

Reviewer 2 indicate “Fig 3, Dermascan test: The unit is given as % as described in materials and methods but the diagramm shows 0,2..etc. The unit is missing at y-axis. For all parameters it would be helpfull to have a reference value (if available from literature or databases and applicable). For Fig.4 an increase of 37.6 % seems very high, as written in the text. The bars show differences between 45 to 60 PU. Please explain the numbers in the text”. The authors appreciate the reviewer's comment. Figure 3 has been modified to make it clearer that the parameter evaluated is the proportion of non-echogenic tissue (%), and this has been clarified in the text. In relation to figure 4 we have reviewed the data and the analysis carried out, and we apologize for the error. There is not an increase of 37.6%, but rather 22.6%.

On the other hand, to the extent possible, normality values ​​have been sought for most of the parameters evaluated. These normality values ​​have preferably been added in a specific section of the manuscript, along with the analysis of differences between men and women for these parameters at day 42 (D42). New bibliographic citations have been added.

Reviewer 2 indicate: “Line 352: „Both the medical practitioner and the subjects who received the CBL treatment indicate that there is an apparent improvement in 50-62.5% of the cases, and that this improvement is important between 20-30% of the cases.“ Please rephrase this sentence as the improvement refer either to the practioner or the patient (i.e. 20 and 30 % judged, respectively). As it is written now, this is not clear to the reader and might be misleading. How do the authors calculate 70-92,5 % from table 5? This is not clear and needs to be explained or rephrased”. The authors appreciate the reviewer's comment. The wording has been reformulated to avoid confusion for readers.

Reviewer 2 indicate: “Discussion: Carefully check the use of HMW-HA. It seems to got mixed up in the discussion”. The reviewer's comment is appreciated. All discussion text has been standardized to HMW-HA.

Reviewer 2 indicate: “Line 532: „well known that ultraviolet radiation also generates oxygen free radicals in“  The authors use the term ROS in the following sentences. Perhaps they could introduce ROS instead of ‚oxygen free radical‘ and use it throughout”. The reviewer's comment is appreciated, a phrase on oxygen free radicals and reactive oxygen species has been included in the discussion, with bibliographical citation.

Reviewer 2 indicate “line 620: Luminosity depends on luminosity…... Please rephrase this sentence”. The reviewer's comment is appreciated. The sentence has been modified.

The modifications made based on the reviewers' suggestions and comments have been marked in yellow in the modified manuscript. New information based on these suggestions has been marked in green in the modified manuscript.

Round 2

Reviewer 1 Report

the manuscript was revised according to suggestions

as above